# Comparison of Hot Cracking Susceptibility of TIG and Laser Beam Welded Alloy 718 by Varestraint Testing

**Pedro Alvarez** [1,*]**, Lexuri Vázquez** [1]**, Noelia Ruiz** [2]**, Pedro Rodríguez** [2] **, Ana Magaña** [3]**, Andrea Niklas** [3] **and Fernando Santos** [3]

1   IK4-LORTEK Technological Centre, Arranomendia kalea 4A, 20240 Ordizia, Spain
2   Alfa Investigación, Desarrollo E Innovación, Torrekua 3, 20600 Eibar, Spain
3   IK4-AZTERLAN, Aliendalde Auzunea 6, 48200 Durango, Spain
*   Correspondence: palvarez@lortek.es; Tel.: +34-943-88-23-03

**Abstract:** Reduced hot cracking susceptibility is essential to ensure the flawless manufacturing of nickel superalloys typically employed in welded aircraft engine structures. The hot cracking of precipitation strengthened alloy 718 mainly depends on chemical composition and microstructure resulting from the thermal story. Alloy 718 is usually welded in a solution annealed state. However, even with this thermal treatment, cracks can be induced during standard industrial manufacturing conditions, leading to costly and time-consuming reworking. In this work, the cracking susceptibility of wrought and investment casting alloy 718 is studied by the Varestraint test. The test is performed while applying different welding conditions, i.e., continuous tungsten inert gas (TIG), low frequency pulsed TIG, continuous laser beam welding (LBW) and pulsed LBW. Welding parameters are selected for each welding technology in order to meet the welding quality criteria requested for targeted aeronautical applications, that is, full penetration, minimum cross-sectional welding width and reduced overhang and underfill. Results show that the hot cracking susceptibility of LBW samples determined by the Varestraint test is enhanced due to extended center line hot cracking, resulting in a fish-bone like cracking pattern. On the contrary, the minor effect of material source (wrought or casting), grain size and pulsation is observed. In fact, casting samples with a 30 times coarser grain size have shown better performance than wrought material.

**Keywords:** hot cracking; Ni superalloys; alloy 718; varestraint test; TIG welding; laser beam welding; grain size

## 1. Introduction

High temperature Ni-base superalloys are broadly employed for the manufacturing of high responsibility hot section parts of aircraft turbines. Precipitation-strengthened Ni superalloys are suitable to work at high temperatures (above 700 °C), under high structural loads and corrosive conditions due to their outstanding tensile, fatigue, creep and oxidation resistant properties [1].

The earliest Ni superalloys were strengthened by the precipitation of $\gamma'$ (Ni3Al, Ni3Ti and Ni3(Ti, Al)) but they underwent severe strain age cracking (SAC) after welding. Additionally, the post weld heat treatment limited their manufacturability. In 1963, Inconel 718 alloy was developed, which added Nb into the matrix which forms metastable $\gamma''$ precipitates of $Ni_3Nb$. The precipitation kinetics of $\gamma''$ are slower compared to $\gamma'$ which contributes to the improvement of weldability and resistance to SAC [1]. Besides $\gamma''$, secondary and intermetallic phases such as NbC and Laves phases can also be found in alloy 718 castings and welds due to the segregation of chemical elements during solidification [2].

Despite modern alloy 718 being considered readily weldable due to the tight chemical composition control and dedicated thermal treatments, hot cracking problems still persist in industrial manufacturing. This leads to expensive and lengthy repairing rework. Alloy 718 is usually welded in a solution annealed state, since numerous studies have revealed that its hot cracking susceptibility is much lower in comparison to the ageing state [3–8]. Even in this thermal condition, cracks can still appear in the heat affected zone (HAZ) due to liquation cracking and in the fusion zone (FZ) due to solidification cracking. HAZ liquation cracking is caused primarily by the constitutional liquation of the Laves phase and carbide precipitates in the grain boundaries, and the liquation of the δ phase. The cracking also seems to be promoted by coarse microstructures in the HAZ [9–12]. On the other hand, solidification cracking can occur due to the formation of γ/NbC and γ/Laves constituents in the interdendritic zones as a result of the segregation of chemical elements such as B, C and Zr or impurities. For this reason P and S must be kept as low as possible. Residual stresses, solidification shrinkage and weld pool size also influence the susceptibility to this cracking mechanism [1,13].

Previous research works have thoroughly investigated the influence of chemical composition and microstructure resulting from different thermal treatments on the hot cracking susceptibility of alloy 718 [9,14–18] and alloy 738 [19]. This has led to contradictory conclusions about the influence of grain size on the hot cracking susceptibility of alloy 718 castings. Whereas some authors have shown that coarser base metal grain sizes enhance FZ solidification cracking [17,20,21], others concluded on the reduced cracking susceptibility of samples with coarser grains [15].

Aside from chemical composition and microstructure, welding technology and parameters also influence hot cracking susceptibility. In this sense, different authors have recently reported the benefits of using pulsed current to refine solidification microstructure and reduce the amount of Laves phase and Nb segregations in TIG welds [22,23]. Laser beam welding (LBW) has also been extensively investigated for the manufacturing of alloy 718 parts [20,24–26]. Compared to arc welding technologies, the high-power density of LBW provides higher depth/width weld bead ratios, lower residual stresses and distortion, lower heat input and faster solidification which all reduce hot cracking susceptibility. However, LBW of alloy 718 can give rise to porosity within FZ and cracking if proper shielding conditions and welding parameters are not employed [27]. Khan et al. [26] reported that both geometry and weld quality criteria requested by the AWS D17.1M:2010 standard can be achieved over a rather wide process window of welding speed and laser power with 1 μm high beam quality laser source in 3.35 and 5 mm thickness alloy 718 samples. However, this window was greatly reduced for getting liquation crack free welds. Moreover, the process window can be shortened if higher welding quality criteria are requested.

In this work, the hot cracking susceptibility of wrought and investment casting alloy 718 is studied by the Varestraint test. The test consists of the application of a bending deformation along the length of the weld [1,16,28,29]. It uses several interchangeable die blocks to produce specific augmented strains that enhance cracking in both FZ and HAZ. The cracks are then measured after testing, and the sum of them gives the total crack length (TCL). The curve of TCL versus augmented strain gives a qualitative outlook of material weldability.

The Varestraint test provides a good relative measurement of hot solidification cracking susceptibility [1,28]. Liquation and solidification cracking, both contributing to hot solidification, can be studied separately in simple geometries and without sample preparation. This test allows the comparison and ranking of cracking susceptibility of alloys (base metals and filler metals) besides the analysis of the influence of welding conditions on resulting cracking.

The novelty of the current work lies in the use of different welding technologies, i.e., continuous TIG, low frequency pulsed TIG, continuous LBW and pulsed LBW for the performance of Varestraint tests of wrought and investment casting alloy 718. These technologies are widely employed by industrial companies for assembly tasks. The influence of these welding conditions on Varestraint test results of alloy 718 has not been extensively investigated up until now, as there is a small quantity of equipment in the world that can implement them [30].

## 2. Materials and Methods

150 mm (length) × 50 mm (width) × 3.2 mm (thickness) samples of wrought and investment casting alloy 718 alloy were employed to carry out the Varestraint tests. The investment casting alloys were casted at ALFA INVESTIGACIÓN, DESARROLLO E INNOVACIÓN facilities under vacuum conditions in plates that were originally of 10 mm thickness. Two testing samples were machined out of each casting plate by electric discharge machining from the midsection and removing the external surfaces in both samples faces.

The chemical compositions of tested samples are included in Table 1. The wrought material fulfilled the standard AMS 5596 and it was in the solution annealed state (995 °C ± 10 °C). The investment casting material was subjected to HIP treatment (at 1123 °C, 4 h), post-HIP treatment (at 1044 °C, 1 h) and solution annealing treatment (at 955 °C, 1 h) before welding.

**Table 1.** Chemical composition of wrought and investment casting alloy 718 in weight %.

| Material | C | Si | S | Co | Cr | Mo | Cu | Mn | P | Ti | Al | B | Ni | Nb | V | W | Fe | Mg |
|---|---|---|---|---|---|---|---|---|---|---|---|---|---|---|---|---|---|---|
| Wrought | 0.052 | 0.09 | 0.002 | 0.46 | 18.4 | 3.09 | 0.05 | 0.23 | 0.009 | 1.00 | 0.54 | 0.004 | 52.2 | 5.03 | - | - | 18.8 | - |
| Investment casting | 0.046 | 0.14 | 0.002 | - | 17.7 | 3.20 | - | 0.04 | 0.004 | 0.93 | 0.47 | - | 54.6 | 5.11 | 0.04 | 0.24 | 18.1 | 0.12 |

The Varestraint tests were completed in a testing device that was fully designed and manufactured at LORTEK. This device allows the application of both TIG and LBW during the test (Figure 1). In this study autogenous welds were performed by melting testing base metals and without adding any filler metal. Different augmented strains were applied using different die block radii (varying from 20 to 320 mm) according to the following equation:

$$\varepsilon = \frac{t}{2 \cdot R} \times 100, \tag{1}$$

where $\varepsilon$ is the augmented strain in %, $t$ is the thickness of the sample in mm and $R$ is the radio of the die block in mm. This led to $\varepsilon$ in the range of 0.5–8%. Two expendable support plates of 304 stainless steel were positioned in both sides of the testing samples to avoid kinking.

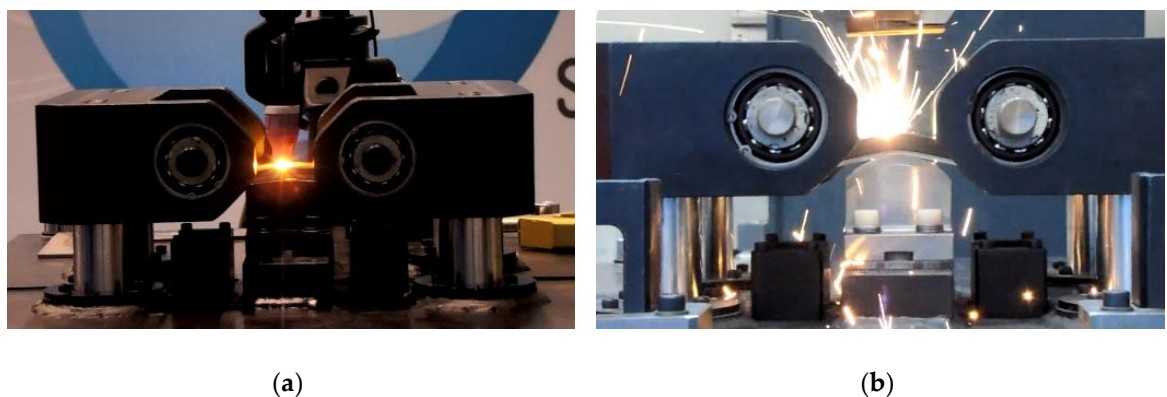

(**a**)          (**b**)

**Figure 1.** The Varestraint testing device showing the completion of a test with: (**a**) TIG; (**b**) LBW.

The performance of the test bench was fully validated [29] and complies with the general requirements of ISO/TR 17643-1 "Destructive tests on welds in metallic materials-Hot cracking tests for weldments-Arc welding processes-Part 3: Externally loaded tests". The system is designed to work both with arc welding and laser beam welding equipment and different testing parameters (stroke rate, welding rate, synchronization set-ups, etc.) can be selected. Nevertheless, stroke rates of 10 and 150 mm/s were selected for the TIG and LBW Varestraint tests, respectively.

The Varestraint tests were completed by TIG welding with the TIGSpeed 352 Synergic RCHW model from the EWM brand. This power source allows for this to work both in continuous mode

and with selected frequency pulsation. All the TIG welding trials were performed at 1 mm/s welding speed (Ws), with 2 mm arc length and 30° electrode tip of 2.4 mm diameter. The electrode was 2% Thoriated tungsten electrode (AWS EWTh-2). The Ar gas blown through the welding torch at 15 l/min was enough to provide good surface condition to welds.

LBW was applied during the complementary Varestraint test by TRUDISK 6002 disk laser from TRUMPF company (Ditzingen, Germany). The laser beam was guided through a 400 µm diameter fiber to TRUMPF BEO D70 laser welding head (200 mm focus length and 200 mm collimation length). LBW Varestraint tests were completed at 0.5 m/min Ws, where a 0.8 mm diameter spot size was used. The tests were performed in a closed envelope filled by Ar gas to avoid surface oxidation and provide shielding conditions.

A LEICA DVM6 digital microscope was employed to measure TCL after the Varestraint test at high magnification (150×). This was previously reported as the optimum measuring condition to reduce standard deviations and the scattering of the experimental TCL measurements [29].

Base materials and welds were characterized by optical microcopy and scanning electron microscopy (SEM). An Energy Dispersive X-ray spectroscopy (EDX) analysis was conducted to determine the local chemical composition of precipitates and phases. A quantitative analysis of carbides as secondary phases was performed.

A metallography analysis of base material and cross-sections of welds was carried out by cutting and mounting samples and subsequently etching by Kalling's 2 reagent (2 g CuCl2, 49 mL HCl, 40–80 mL ethanol). Weld bead dimensions, including minimum weld width (Wm), weld width on face side (B), weld width on root side (C), face underfill on face side, root overhang (Ch) and weld bead angle ($\alpha$) were measured. LBW parameters were selected to achieve Wm above 1.5 mm. These dimensions are included in Figure 2.

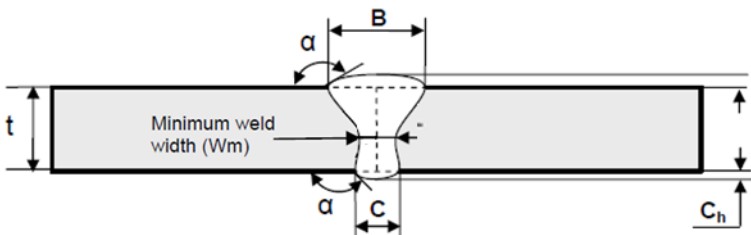

**Figure 2.** Dimensions of welds.

## 3. Results

### 3.1. Base Material Microstructure

The main microstructural features of the wrought and investment casting alloy 718 samples are depicted in Table 2 and shown in Figure 3. It is worth mentioning that wrought samples had a grain size below 30 microns, whereas the grain size of investment casting samples was around 1 mm. That means that grains of casted samples were more than 30 times coarser. In both cases, Varestraint samples had a homogenous microstructure along the 3.2 mm thickness. There were no differences in the percentage of Nb carbides and in both materials small amount of TiNb carbonitrides (TiNbCN) was detected (black precipitates in Figure 3). Laves phases were not observed in any of the materials, but investment casting samples had traces of needle like delta ($\delta$) phase and Mo sulphide which are highlighted by arrows and a circle in Figure 4b, respectively. EDX analysis confirmed the nature of each observed phase and precipitate.

**Table 2.** Microstructural features of wrought and investment casting alloy 718 samples.

| Material | Grain Size (μm) | Morphology of Grains | % of MC Carbides | Other Phases |
|---|---|---|---|---|
| Wrought | 22.5 ± 6.5 16–25 [1] | Equiaxed grains with twins | 0.76% | No Laves phase TiNbCN |
| Investment casting (10 mm as-cast thickness) | 1035 ± 275 760–1310 [1] | Almost equiaxed with aspect ratio 1.2–1.5 | 0.70% | No Laves phases TiNbCN δ phase in grain boundaries |

[1] Maximum and minimum mean length and width of grains measured in one micrograph.

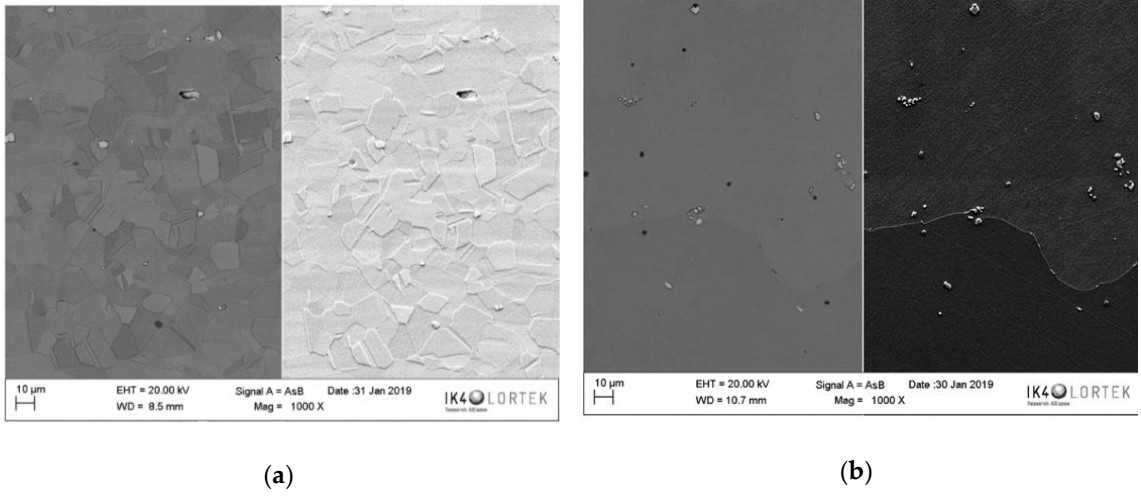

(**a**)                (**b**)

**Figure 3.** SEM images of (**a**) wrought and (**b**) investment casting samples.

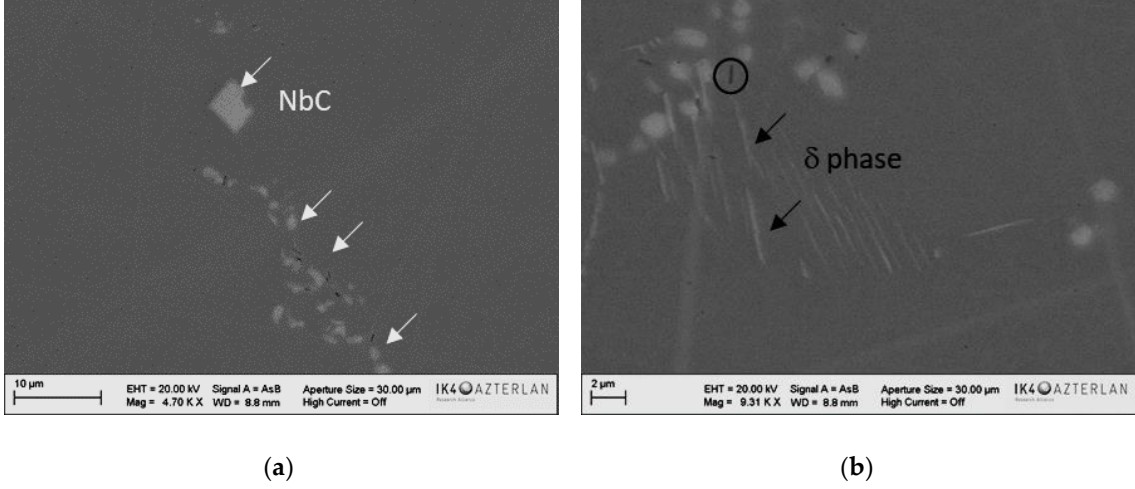

(**a**)                (**b**)

**Figure 4.** (**a**) NbC precipitates; (**b**) traces of δ phase and Mo sulphide (encircled) observed in investment casting alloy 718.

*3.2. Welding Parameters and Microstructure*

3.2.1. TIG Welding

Initially, bead on plate trials were performed in order to define TIG welding parameters to ensure full penetration and reduced underfill and overhang. Continuous and low frequency (LF) pulsed TIG welding curves were investigated. As it is shown in Table 3, the heat input was increased 13% for the pulsed curve with the purpose of getting similar weld penetration and weld morphology (Figure 5).

The weld bead width of face side was approximately 7.5 mm, whereas at the root side it was slightly higher than 4 mm. Having comparable weld bead dimensions, the surface look of the LF pulsed welds was completely different, since some marked ripples (semicircles) were observed due to the pulsation effect.

**Table 3.** Selected welding parameters for continuous and LF pulsed TIG.

| TIG Curve | Ws (mm/s) | $I_{peak}$ [1] (A) | $I_{base}$ [1] (A) | Frequency (Hz) | Voltage (V) | Heat Input (J/mm) |
|---|---|---|---|---|---|---|
| Continuous | 1 | 70 | 70 | 0 | 9.25 | 647.5 |
| LF pulsed | 1 | 115 | 42 | 2.5 | 9.35 | 734 |

[1] $I_{peak}$ and $I_{base}$ stand for maximum and minimum intensities of the pulsed current.

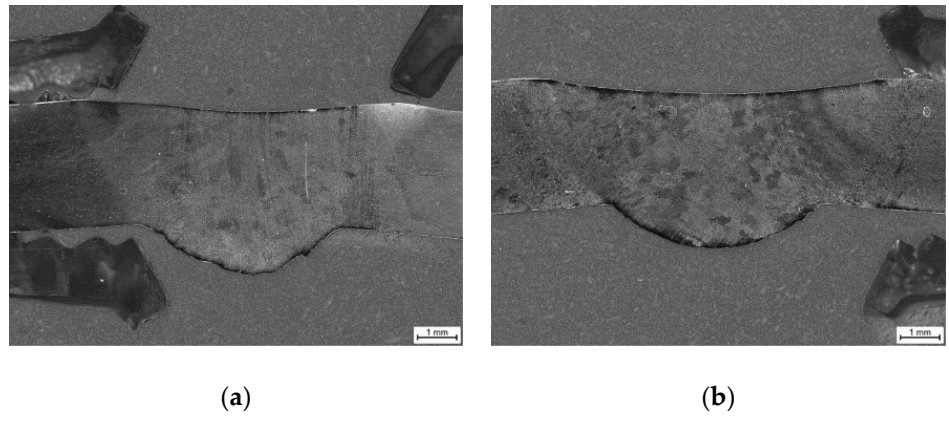

(**a**)　　　　　　　　　　　　　　　　　　　　　(**b**)

**Figure 5.** Cross sections of (**a**) continuous and (**b**) LF pulsed TIG welds.

### 3.2.2. LBW

The influence of continuous and pulsed LBW parameters on welding morphology was also studied. Preliminary tests at different laser powers were performed in order to obtain full penetration in 3.2 mm thickness samples. Initial trials were carried out at 2 m/min WS and 0.4 mm laser spot size. However, these welding conditions led to excessively narrow weld widths, below the aimed 1.5 mm Wm. The reduction of Ws to 0.5 m/min yielded wider welds but caused internal porosity due to entrapped gas. This internal porosity was successfully removed after increasing the spot size to 0.8 mm while keeping previous Ws. The spot was modified by defocusing the laser bead 4 mm in the positive direction. These welding parameters were suitable to get sound welds, to ensure target Wm across sample thickness and meet welding quality criteria which are usually specified for aeronautical applications. Laser power was adjusted for both continuous and pulsed LBW, i.e., rectangular modulated with short pulses lasting 6 ms, to provide full penetration and minimum overhang. Optimum LBW parameters are included in Table 4. In this case, energy and energy density were similar for both conditions.

**Table 4.** Selected welding parameters for continuous and pulsed LBW.

| LBW Curve | Ws (mm/s) | $P_{peak}$ [1] (W) | $P_{base}$ [1] (W) | Frequency (Hz) | Energy (J/mm) | Energy Density (J/mm$^3$) |
|---|---|---|---|---|---|---|
| Continuous | 8.33 | 2300 | 2300 | 0 | 276 | 549 |
| Pulsed | 8.33 | 3166 | 1582 | 83 | 285 | 567 |

[1] $P_{peak}$ and $P_{base}$ stand for maximum and minimum power of the pulsed curve.

Weld bead dimensions are included in Figure 6. It is worth noting that the weld bead width of face side (between 4.8 and 4.9 mm) and root side (approx. 2.11 and 1.64 mm) for both LBW curves were narrower in comparison with TIG weld beads.

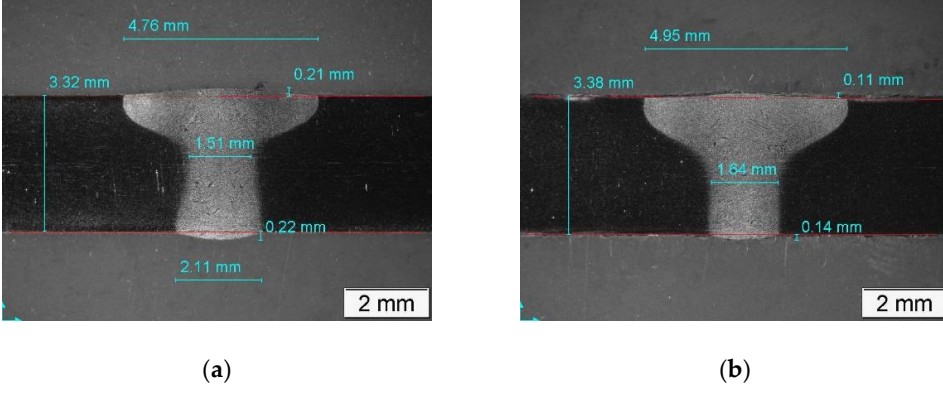

(**a**)                                                    (**b**)

**Figure 6.** Cross sections of (**a**) continuous, (**b**) pulsed LBW welds.

### 3.3. Varestraint Test Results

#### 3.3.1. Influence of Welding Parameters on Cracking Susceptibility of Wrought Alloy 718

Once welding parameters were defined, Varestraint tests were carried out in wrought samples. Four TCL versus augmented strain curves were obtained applying continuous TIG, low frequency pulsed TIG, continuous LBW and pulsed LBW. Corresponding curves are brought together in Figure 7. Each individual TCL value was taken from an average of three samples. Errors, in terms of standard deviation for each measurement, are included in the figures. The standard deviation of TIG welded samples was so small (below 0.15 mm), that it is not distinguished in Figure 7 curves.

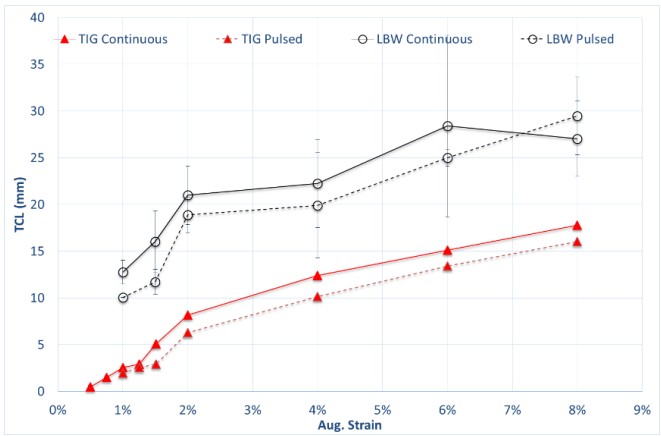

**Figure 7.** TCL versus augmented strain ($\varepsilon$) curve resulting from the Varestraint test and showing the cracking response of wrought alloy 718 under different welding conditions.

The results show that for both welding technologies, pulsation has a reduced but beneficial effect on the drop of TCL measured for every $\varepsilon$ and therefore, on cracking susceptibility. As it has been observed in Figure 8 for TIG welding, an advantageous effect of pulsation relies on a reduction of HAZ cracking susceptibility at low augmented strains and reduced FZ cracking susceptibility at high augmented strains. Note that individual contributions to TCL are comparable for the same augmented strains and remain below 10 mm at the maximum tested augmented strain (8% $\varepsilon$).

Regarding LBW, both curves are laid on top of TIG ones (Figure 7). This means that hot cracking susceptibility is enhanced under described LBW conditions. However, a separate analysis of HAZ and FZ cracking response shows that whereas HAZ liquation cracking is reduced in comparison to TIG, i.e., maximum TCL contribution below 4 mm (Figure 9a), cracks measured in FZ reach 25 mm at 8% $\varepsilon$ (Figure 9b). Consequently, one can easily conclude that the enhancement of cracking susceptibility is directly linked with the much higher solidification cracking tendency. As stated before, this does not seem to be greatly affected by LBW pulsation.

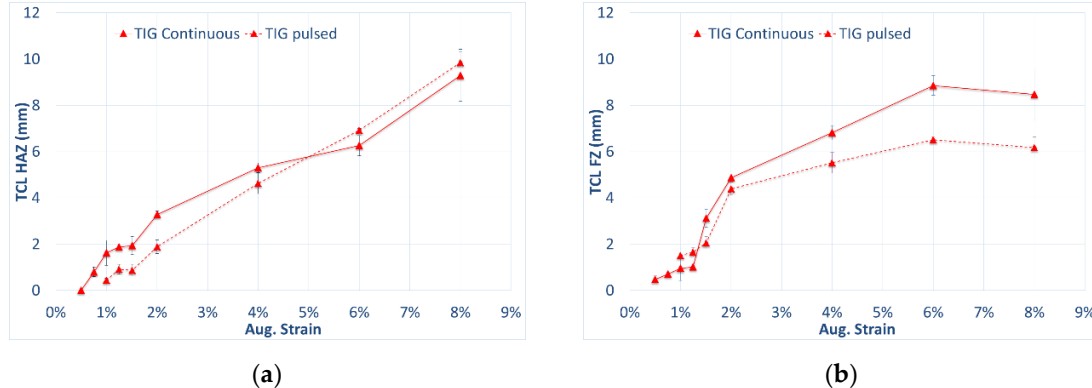

**Figure 8.** (**a**) HAZ and (**b**) FZ cracking response of wrought alloy 718 under different TIG welding conditions.

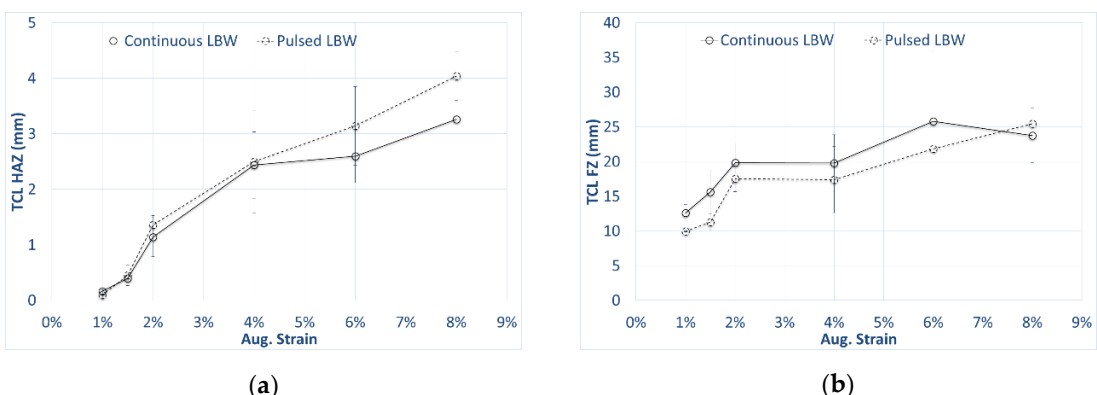

**Figure 9.** (**a**) HAZ and (**b**) FZ cracking response of wrought alloy 718 under different LBW welding conditions.

3.3.2. Comparison between Wrought and Investment Casting Samples

Figure 10 compares the cracking response of wrought and investment casting samples tested in similar continuous TIG and LBW conditions. In both cases, the cracking susceptibility of cast samples was significantly lower than wrought counterparts. Note that the base metal grain size was more than 30 times coarser in the cast material. Therefore, current results show that coarser grain sizes in alloy 718 is not necessarily related to higher hot cracking susceptibility. This is something that has been traditionally considered as a general rule until the publication of divergent findings [15].

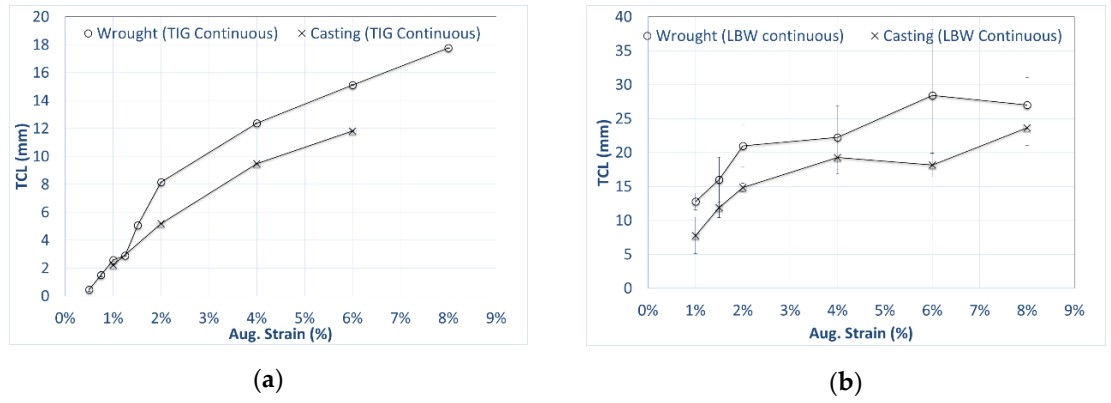

**Figure 10.** Comparison of cracking response of wrought and cast alloy 718 samples tested under (**a**) continuous TIG and (**b**) continuous LBW conditions.

## 4. Discussion

Reported results show an enhanced hot cracking susceptibility of LBW alloy 718 in comparison with TIG. This was observed for both wrought and casting samples that were machined out from the same plates and therefore any influence of chemical composition on these results is discarded. Note that these results and conclusions were obtained employing selected welding parameters and heat inputs which lead to welds with tight industrial quality requirements. Particularly, full penetration, minimum cross-sectional welding width (Wm) above 1.5 mm and reduced overhang and underfill were the critical weld quality requirements. One cannot conclude that the relative ranking between TIG and LBW will be the same if welding parameters are changed and the requirement of Wm is released.

This conclusion can be considered a ground-breaking finding, since to the author's best knowledge, comparable Varestraint studies with LBW have not been previously published for alloy 718. Chun et al. [31] reported about solidification cracking behavior in LBW austenitic stainless steels. They applied the transverse-Varestraint test while welding with fiber laser. They concluded that solidification cracking tendency cannot be simply mitigated through application of the LBW process. Recently, Raza et al. [30] have conducted Varestraint weldability testing of additive manufactured alloy 718. Samples were manufactured by selective laser melting (SLM), a powder bed fusion technology employing laser source for the selective fusion of pre-deposited thin powder layers. However, they carried out the Varestraint tests using TIG welding equipment. Therefore, they were able to conclude on the cracking susceptibility of the manufactured parts, but not on cracking behavior during solidification of the laser assisted SLM process.

As it has been previously introduced, enhancement of the alloy 718 cracking susceptibility by LBW is related to prominent FZ solidification cracking (Figure 9). This behavior can be explained by looking at the cracks observed in the face side of tested samples. The cracking pattern of equivalent TIG and LBW Varestraint samples is compared in Figure 11. Both samples show small cracks in the HAZ which are perpendicular to the welding direction. TIG samples had a longer and higher number of cracks in the HAZ, matching with slightly higher HAZ TCL, as observed in Figures 8a and 9a. However, the cracking pattern is completely different in the FZ. The TIG sample displays a limited number of cracks which are perpendicular to the semicircular solidification line. These cracks progress up to the HAZ. In the LBW sample a fish-bone like cracking pattern is observed. This is characterized by long center line cracking, and straight but inclined cracks coming from it and from the enlarged V-shape solidification line. Inclined cracks are confined only in the FZ. Despite only two samples being shown, it must be noted that these cracking patterns were characteristics of every TIG and LBW sample, and therefore, one can conclude that differences in solidification patterns determined the distinctive hot cracking response. Note that the face weld widths are approximately 7.5 and 4.9 mm for TIG and LBW samples. It is worth mentioning again that the elongated V-shape solidification line of LBW was a result of the required energy density to achieve full penetration and Wm above 1.5 mm.

For both TIG and LBW welding, pulsation did not give rise to significant solidification pattern change and cracks in pulsed welding samples looked like continuous ones. As previously mentioned, pulsation promoted the formation of more pronounced ripples and semicircles in the weld bead, especially in TIG samples, but this did not entail any remarkable difference in the resulting number of cracks and TCL. As it is observed in Figure 8a, pulsation enabled a slightly reduced HAZ cracking in TIG samples at low $\varepsilon$. This is in line with the general assessment that pulsation reduced base metal thermal affection. However, as the heat input of pulsed parameters was increased to achieve full penetration in the current study (Table 3), it is very likely that both effects have been compensated. On the contrary, in LBW samples, comparable energy and energy density were employed for continuous and pulsed curves. In this case, the contribution to the TCL of HAZ was so small in comparison with FZ, that potential differences would be insignificant for the hot cracking behavior.

In terms of FZ cracking, it was previously reported that the use of current pulsing refines the fusion zone microstructure, reduces the amount of Laves phase and limits Nb segregations [22,23]. It is considered that these three issues are beneficial to reducing hot cracking. However, current results

do not support any evidence or conclusion along those lines. In fact, in all cases the microstructural analysis of welds revealed a coarse structure with a high amount of Nb segregation and Laves phase in interdendritic regions of FZ. Cracks induced by the Varestraint test were always following these boundaries and especially those with a high concentration of the Laves phase which created a continuous network structure (Figure 12). Again, differences in welded microstructures due to pulsation were not evident, and a comparable response in terms of FZ cracking was observed.

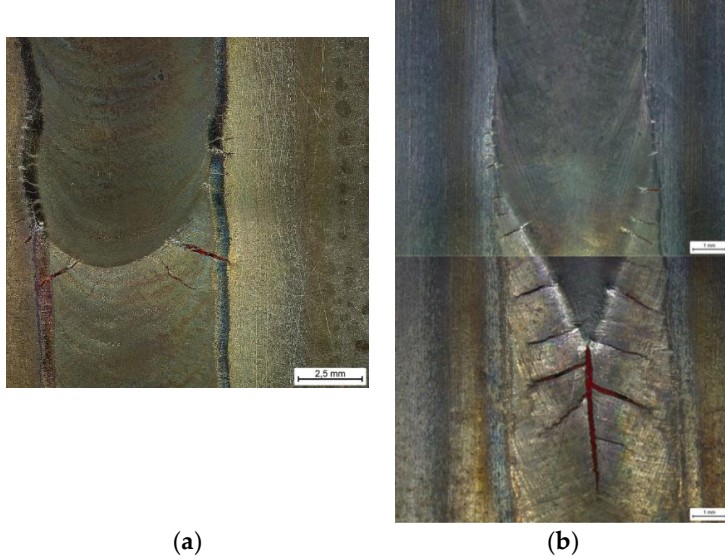

|     (a)     |     (b)     |

**Figure 11.** Aspect of face side of continuous (**a**) TIG and (**b**) LBW Varestraint samples. Both samples were tested at 2% ε.

Nb rich Laves phase is a brittle intermetallic phase of Ni, Cr, Fe2, or Nb, Mo, Ti. As Laves phases diminish the matrix of vital alloying elements, they are favorable sites for crack initiation and facilitate their growth [22,23]. EDX analysis of the phase observed in the middle of Figure 12b confirmed that this phase was composed of 35–41% Ni, 13–14%Cr, 13% Fe, 25–31% Nb, 5% Mo and 1% Ti.

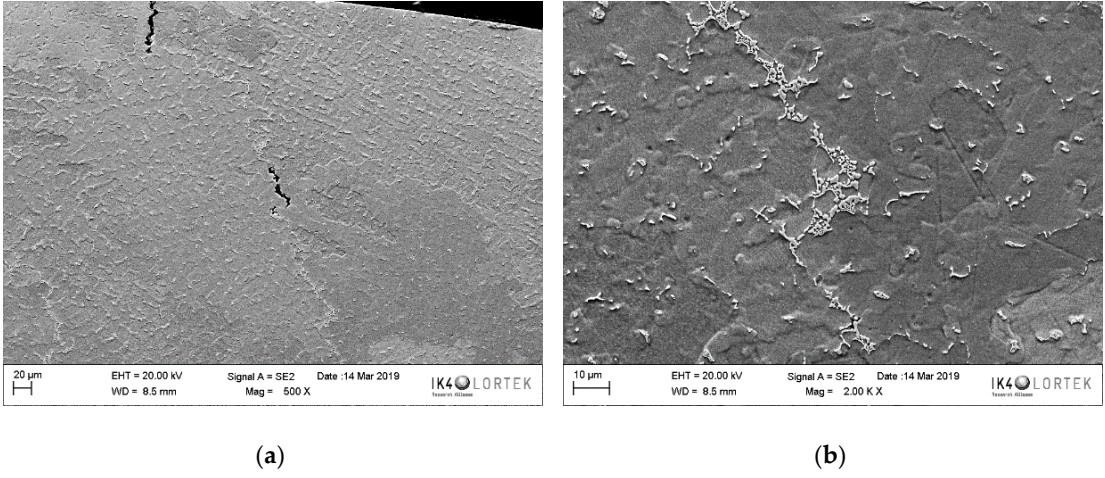

|     (a)     |     (b)     |

**Figure 12.** SEM image of (**a**) FZ cross section of continuous TIG Varestraint sample tested at 2% ε and (**b**) Continuous network of Laves phase is observed in FZ interdendritic areas, and cracks follow this network.

Last but not least, in this study a consistent result has been observed when comparing wrought and investment casting samples. Irrespectively of the welding technology, casting samples always show comparatively reduced cracking susceptibility. This can be considered as an unexpected result,

since casting samples had a much coarser grain size, and this is usually associated to an enhanced cracking trend [1,17,20,21]. Current results do not completely preclude previous investigations, since chemical composition of tested samples were not exactly the same, and the influence of minor elements on hot cracking susceptibility cannot be neglected. The positive outcome from this research work is that thick casting plates of alloy 718 can be even more readily weldable than solution annealed wrought material with fine and homogenous microstructure. Note that in this study, tested samples were machined out of thickness casting plates that were originally 10 mm.

## 5. Conclusions

The main conclusions of the current work can be summarized as:

- The hot cracking susceptibility of alloy 718 determined by the Varestraint test and resulting from applying LBW was higher than that for TIG welding conditions. Welding parameters were thoroughly selected to get sound welds and meet industrial welding quality criteria, particularly in terms of minimum weld width (Wm), underfill, overhang and porosity.
- LBW enhanced hot cracking tendency was directly related to a different solidification pattern (enlarged V-shape) that promotes long center line FZ solidification cracking and fish-bone like cracks.
- Minor effect of material source (wrought versus investment casting), grain size and pulsation was observed. Investment casting with much coarser average grain size (around 1 mm) shown comparatively better cracking behavior than wrought material (grain size below 30 microns). Both base materials had a microstructure free of Laves phases and Nb segregations, but they were observed in the interdendritic boundaries of the welds creating a network and decorating the path of FZ cracks.

**Author Contributions:** Conceptualization, P.R. and F.S.; methodology, P.A., L.V., N.R., P.R., A.M., A.N., F.S.; formal analysis, P.A., L.V., N.R. A.M., A.N.; investigation, P.A., L.V., N.R., P.R., A.M., A.N.; data curation, L.V., N.R., A.M., A.N.; writing—original draft preparation, P.A.; writing—review and editing, L.V, P.R., F.S.; visualization, L.V.; supervision, F.S.; project administration, F.S.; funding acquisition, F.S.

**Funding:** This research has been performed in the framework of HiperTURB project which was funded by Clean Sky 2 Joint Undertaking under the European Union's Horizon 2020 research and innovation program, grant agreement No 755561.

**Acknowledgments:** Bengt Pettersson and Vikström Fredrik from GKN Aerospace company in Trollhättan (Sweden) are gratefully acknowledged for their technical support and fruitful discussions.

**Conflicts of Interest:** The authors declare no conflict of interest.

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
