# Peer review of "Comparison of Hot Cracking Susceptibility of TIG and Laser Beam Welded Alloy 718 by Varestraint Testing"

_metals, doi:10.3390/met9090985_

Round 1

Reviewer 1 Report

Dear authors,

I would like to thank you for submitting a very nice and interesting paper which I really enjoyed reading!

I have some comments that I would like you to attend to:

 the overall language in whole paper need improvement. I suggest you take some help from a native English person for this matter. It is important to get a nice flow when reading as well as to avoid misunderstanding. Alloy 718 is a Ni-Fe based precipitation hardening alloy, please change  The sentence on line 58-59 is a bit misleading when it refers to FZ, please change. Add the respective dwell times for all the heat treatments (line 95-98) What type of TIG electrode was used (line 120-122) How can you be so sure that the small arrows in Fig. 4a are not associated with Laves phase? Fig. 4a and b are a bit blurry. The scatter in TCL for TIG-Varestraint is remarkably low, could you please comment on this and in particular why you think you have succeeded so well in this context? Line 209, what do you mean by "Note that individual contributions to TCL are comparable and below 10 mm at 8% E."? No pulsed welding for the cast samples? Line 298, "(10 mm as-cast thickness)" is misleading since you tested a thickness of 3.2 mm so please rephrase. You need to attend to and discuss the meaning and any possible influence the difference in heat input as well as weld bead width might have on the ranking between TIG and LBW since the difference that you face in the current study is a bit of comparing "apples and pears".

Best regards

Author Response

Point 1:  The overall language in whole paper need improvement. I suggest you take some help from a native English person for this matter. It is important to get a nice flow when reading as well as to avoid misunderstanding 

Response 1: Thank you for the suggestion. We are not English native speakers, but we tried to write a clear paper in plain English. We have considered to ask for professional language editing support (including the service provided by MDPI), but since the rest of 2 reviewers requested only fine/minor spell checking, we made an internal revision and correct the detected spelling mistakes. Some sentences that could be misleading have been written again.

Point 2: Alloy 718 is a Ni-Fe based precipitation hardening alloy, please change

Response 2: We are not sure what do you mean. We use the terms “Ni-base superalloys” and “precipitation-strengthened Ni superalloys” during the general introduction in lines 31-33, to refer not only to Alloy 718. These are usually employed by scientific and industrial community. See for example, the book “Welding metallurgy and weldability of nickel-base alloys”.

Point 3: The sentence on line 58-59 is a bit misleading when it refers to FZ, please change

Response 3: The sentence has been change in new paper version

Old version: Whereas some authors have shown significant detrimental influence of base metal grain size on FZ cracking [17][18][19], others concluded on reduced, but positive effect [15].

New version: Whereas some authors have shown that coarse base metal grain sizes enhance FZ solidification cracking [17][18][19], others concluded on comparable cracking susceptibility of samples with different grain sizes [15].

Point 4: Add the respective dwell times for all the heat treatments (line 95-98)

Response 4: Done

Investment casting material was subjected to HIP treatment (at 1123ºC, 4 h), post-HIP treatment (at 1044ºC, 1 h) and solution annealing treatment (at 955ºC, 1h) before welding.

Point 5: What type of TIG electrode was used (line 120-122)

Response 5: We think this is not relevant for the paper. Anyway, the electrode type was included. “The electrode was 2% Thoriated tungsten electrode (AWS EWTh-2)”.

Point 6: How can you be so sure that the small arrows in Fig. 4a are not associated with Laves phase?

Response 6:  As it is mentioned in the text “EDX analysis confirmed the nature of each observed phase and precipitate”. Particularly for Laves phases, we had local chemical analysis and we identified chemical elements in relative percentages which are close to the Laves phase. “The Laves phase is an intermetallic compound with the A 2 B type structure, where A = Fe, Ni, Cr and B = Nb, Mo, and Si”.

We decided not to include EDX mapping results.

Point 7: Fig. 4a and b are a bit blurry.

Response 7:  Quality of the figures have been improved.

Point 8: The scatter in TCL for TIG-Varestraint is remarkably low, could you please comment on this and in particular why you think you have succeeded so well in this context?

Response 8:  We were also surprised by the low scattering of TCL for TIG-Varestraint samples, but these are the real values we obtained. Scattering values were checked more than once. We can explain them as follows:

High stability and repeatability of our Varestraint testing system. Note that we are using a high-speed and high-accuracy electrovalve with minimum response time. Crack measuring procedure. The operator measured the cracks at high magnification, with improved lighting system and using straight segments instead of tortuous zig-zag lines. Performance of operator. The performance of the operator was brilliant and we use blind tests to demonstrate this. Other operator showed more scattering measuring selected samples.

Point 9: Line 209, what do you mean by "Note that individual contributions to TCL are comparable and below 10 mm at 8% E."

Response 9: We mean that individual TCL measurements for HAZ and FZ cracking are comparable for the same augmented strain levels and remain below 10 mm at the maximum tested augmented strain. For the LBW samples, TCL HAZ is between 1-4 mm, whereas TCL FZ is 10-25 mm.

This sentence was rewritten:

“Note that individual contributions to TCL are comparable for the same augmented strains and remain below 10 mm at the maximum tested augmented strain (8% E).E

Point 10: No pulsed welding for the cast samples?

Response 10: Unfortunately, we had not the chance to test casting samples in pulsed conditions due to limited number of available samples. Since minor influence of pulsation was observed, we decided to concentrated on continuous welding curves.

Point 11: Line 298, "(10 mm as-cast thickness)" is misleading since you tested a thickness of 3.2 mm so please rephrase.

Response 11: The sentence has been change in new paper version

Old version: The positive outcome from this research work is that thick casting samples of alloy 718 (10 mm as-cast thickness) can be even more readily weldable than solution annealed wrought material with 1 and homogenous microstructure.

New version: “The positive outcome from this research work is that thick casting plates of alloy 718 can be even more readily weldable than solution annealed wrought material with fine and homogenous microstructure. Note that in this study tested samples were machined out of originally 10 mm thickness casting plates.”

Point 12: You need to attend to and discuss the meaning and any possible influence the difference in heat input as well as weld bead width might have on the ranking between TIG and LBW since the difference that you face in the current study is a bit of comparing "apples and pears".

Response 12: You are right, this is a critical issue affecting to the conclusions of our work and we are pretty sure that the ranking between TIG and LBW can be different depending on the welding parameters.

The selection of the welding parameters was not arbitrary, but supported by industrial standards. Our results and conclusions can only be applicable to those industrial welding conditions. We have now highlighted this in the discussion.

Reviewer 2 Report

Thanks for a well written manuscript and an extensive experimental work. Please consider the following suggestions for improvement.

Lin 177-185: The differences in laser parameters are poorly presented. How is the spot size changed? Chasnger of focus position? To positive or negative direction? How does this affect the intensity - also relative to tig-process. How is these numbers relative to sheet thickness? 

What is the actual laser peak power (not the theoretical one, but the peak power including overshoot. What is the duty cycle? These numbers highly affects the tendency of vaporisation and reduction of chemical componentents across the weld profile. Please relate finding to this.

Line 216: How is this "easily concluded"? Please justify and relate to other litterature.

show figures with same scaling, e.g. fig 11 contains very different scaling.

Why is there a need for energy inpout for pulksed TIG relative to CW TIG, and why is energy input to LBW different. What influence can these different energy inputs have relative to TLC and chemical/microstructual differences. This is insuficiently discussed.

Author Response

Point 1:  Lin 177-185: The differences in laser parameters are poorly presented. How is the spot size changed? Chasnger of focus position? To positive or negative direction? How does this affect the intensity - also relative to tig-process. How is these numbers relative to sheet thickness? 

Response 1: The spot was modified by defocusing the laser bead 4 mm in the positive direction to shift from 0.4 to 0.8 mm diameter. This has been included in new version.

In the case of LBW, defocusing affects the energy density and distribution (laser power will be always the same). When increasing spot size, the same energy is distributed within a higher area. This results in different weld cross-sections (different penetration and weld morphology).

In the case of TIG, we used always the same arc length (2 mm). This was achieved by touching with the tip of the electrode the surface of the testing sample and moving 2 mm away from the sheet top surface. If this is not controlled, the voltage can change and consequently, the heat input.

According to our knowledge, TIG heat input and LBW energy and energy density values are typical one for 3.2 mm thickness Alloy 718 sheets. Nevertheless, note that these values were obtained after welding parameter optimization process to get required industrial welding quality requirements.

Point 2: What is the actual laser peak power (not the theoretical one, but the peak power including overshoot. What is the duty cycle? These numbers highly affects the tendency of vaporisation and reduction of chemical componentents across the weld profile. Please relate finding to this.

Response 2: In our laser, we introduce a rectangular modulation to chop continuous laser curve in a controlled way. This is an experimental methodology that has been previously employed by other authors (See for example T. Y. Kuo and S. L. Jeng, “Porosity reduction in Nd – YAG laser welding of stainless steel and inconel alloy by using a pulsed wave,” J. Phys. D Appl. phusics, vol. 38, no. 5, pp. 722–728, 2005, DOI:10.1088/0022-3727/38/5/009.)).Therefore, we do not expect significant overshooting.

Duty cycle is 50% and each pulse lasts 6 ms.

We have not found references about vaporisation when using this particular LBW setup.

Point 3: Line 216: How is this "easily concluded"? Please justify and relate to other litterature.

Response 3: FZ cracking is the same as solidification cracking. The latter refers to the mechanism and the former to the location of the cracks. Since HAZ TCL is 1-4 mm and FZ TCL is 10-25 mm, solidification cracking is the cracking mechanism which is mostly contributing to the cracking susceptibility. We have not found any previous reference regarding individual contributions to cracking in LBW samples.

Point 4: show figures with same scaling, e.g. fig 11 contains very different scaling.

Response 4: Image sizes and scales are different because the size of the solidification lines are different. Image size have been selected to compare face weld widths with the same magnification. As it is included in the text, “Note that face weld widths are approximately 7.5 and 5.2 mm for TIG and LBW samples”

Point 5: Why is there a need for energy inpout for pulksed TIG relative to CW TIG, and why is energy input to LBW different. What influence can these different energy inputs have relative to TLC and chemical/microstructual differences. This is insuficiently discussed.

Response 5: The need relies on the welding quality requirements (full penetration, minimum weld witdh 1.5 mm, reduced overhang and underfill). Discussion was adapted to highlight that the selection of the welding parameters was not arbitrary, but supported by industrial standards. Our results and conclusions can only be applicable to those industrial welding conditions.

Reviewer 3 Report

Dear Authors,

I have recived fo revision your manuscrupt titled: "Comparison of hot cracking susceptibility of TIG and laser beam welded alloy 718 by Varestraint testing".

In my opinion there are some informations missing in the manuscript, which have to be writed. I propose major revision.

General remarks:

- Now, only 4 references have been published in last 3 years. In my opinoin it isn't enough. I propose to add some positions. You should also add some positions from Metals journal.

Introduction:

In my opinion, you should exceed some informations in this section.

In lines 55-60, you have written "Previous research works have thoroughly investigated influence of chemical composition and microstructure resulting from different thermal treatments on hot cracking susceptibility of alloy 718 56 [9][14][15][16][17]."

There isn't any info what does i"influence" mean. How? What changed due to changes in presented factors? This paragraph have to be rebulid. Informations should be clear for readers.

You have to change the writing in brackets. Should be eg.:
- line 46 - [3-8]
- line 50 - [9-12]
- line 54 - [1,13]
etc.

Materials and Methods

- table 1 - You have presented the chemical composition of used material. What type of chemical analysis have been done?
- line 120 - "All the TIG welding trials" - wht does it mean "all", how many specimens have you prepared?
- in my opinion you should cleaarly describe why you have used the Varestraint test. Its advantages in comparation to other test.

Results:
I propose to connect "results" and "discussion" part. It will be easier for potential readers, if they can find discussion near the results of each test.|

Conclusions:

Are clear.

Author Response

Point 1:  Now, only 4 references have been published in last 3 years. In my opinoin it isn't enough. I propose to add some positions. You should also add some positions from Metals journal

Response 1: We have updated our references related to the topic and included two additional references from Metals journal.

[29]        K.-C. Chen, T.-C. Chen, R.-K. Shiue, and L.-W. Tsay, “Liquation cracking in the heat-affected zone of IN738 superalloy weld,” Metals (Basel)., vol. 8, no. 6, p. 387, 2018.

[30]        T. Raza, K. Hurtig, G. Asala, J. Andersson, L.-E. Svensson, and O. A. Ojo, “Influence of Heat Treatments on Heat Affected Zone Cracking of Gas Tungsten Arc Welded Additive Manufactured Alloy 718,” Metals (Basel)., vol. 9, no. 8, p. 881, 2019.

We discarded other recent references because they are not totally related with the main topic of the paper.

Point 2: In my opinion, you should exceed some informations in this section

Response 2: We consider this comment a suggestion. Introduction has 7 paragraphs and it is less than one and a half page long. We think current Introduction content is appropriate for the aimed scientific paper. Similar suggestions were not given by the rest of reviewers.

Point 3: In lines 55-60, you have written "Previous research works have thoroughly investigated influence of chemical composition and microstructure resulting from different thermal treatments on hot cracking susceptibility of alloy 718 56 [9][14][15][16][17]."

There isn't any info what does i"influence" mean. How? What changed due to changes in presented factors? This paragraph have to be rebulid. Informations should be clear for readers.

Response 3: This paragraph has been rewritten.

“Previous research works have thoroughly investigated influence of chemical composition and microstructure resulting from different thermal treatments on hot cracking susceptibility of alloy 718 [9][14][15][16][17]. This has led to contradictory conclusions about the influence of grain size on hot cracking susceptibility of alloy 718 castings. Whereas some authors have shown that coarser base metal grain sizes enhance FZ solidification cracking [17][18][19], others concluded on reduced cracking susceptibility of samples with coarser grains [15]”.

Point 4: You have to change the writing in brackets. Should be eg.:

- line 46 - [3-8]

- line 50 - [9-12]

- line 54 - [1,13]

etc.

Response 4: We used Mendeley pluging to insert and track references. We can edit the reference style, but cannot do consolidate references into one bracket. This is something that we can do during final editing.

Point 5: - table 1 - You have presented the chemical composition of used material. What type of chemical analysis have been done?

Response 5: We did not perform chemical analysis by ourselves. We included chemical composition of wrought material given in the material certificate (as standard AMS 5596) and the chemical composition of the ingots employed for the casting.

Point 6: -  line 120 - "All the TIG welding trials" - wht does it mean "all", how many specimens have you prepared?

Response 6: By this sentence we tried to point out that we used always the same welding speed, arc length and electrode in our TIG Varestraint tests. 3 samples were tested for each augmented strain level (tests were carried out at 9 different levels) using both pulsed and continuous welding curves.

Point 7:  - in my opinion you should cleaarly describe why you have used the Varestraint test. Its advantages in comparation to other test.

Response 7: Varestraint tests provides a good relative measurement of hot solidification cracking susceptibility. Liquation and solidification cracking, both of them contributing to hot solidification, can be studied separately in simple geometries and without sample preparation. This test allows to compare and rank cracking susceptibility of alloys (base metals and filler metals) and analyse influence of welding conditions on cracking. These advantages have been included in the new version.

Point 8:  I propose to connect "results" and "discussion" part. It will be easier for potential readers, if they can find discussion near the results of each test.|

Response 8: We consider this comment a suggestion. We prefer to keep current structure which is usually requested for scientific papers. In our opinion, Results sections must describe the results that have been obtained and Discussion must explain these results linking with underpinning mechanism, hypothesis and previous findings (references). We think that the quality of the paper is higher keeping both sections as they are now, as far as good traceability between figures and tables is provided.

Round 2

Reviewer 3 Report

Dear Authors,

You have replied properly to the most of questions and comments that I
pointed out in the first round of review. Your efforts are appreciated.

I have prepared comments to your responses:

Response 1:
DWe have updated our references related to the topic and included two additional references from Metals journal.

[29]        K.-C. Chen, T.-C. Chen, R.-K. Shiue, and L.-W. Tsay, “Liquation cracking in the heat-affected zone of IN738 superalloy weld,” Metals (Basel)., vol. 8, no. 6, p. 387, 2018.

[30]        T. Raza, K. Hurtig, G. Asala, J. Andersson, L.-E. Svensson, and O. A. Ojo, “Influence of Heat Treatments on Heat Affected Zone Cracking of Gas Tungsten Arc Welded Additive Manufactured Alloy 718,” Metals (Basel)., vol. 9, no. 8, p. 881, 2019.

We discarded other recent references because they are not totally related with the main topic of the paper.

In this state, the manuscript includes more of latest published articles and looks better.

Response 2: We consider this comment a suggestion. Introduction has 7 paragraphs and it is less than one and a half page long. We think current Introduction content is appropriate for the aimed scientific paper. Similar suggestions were not given by the rest of reviewers.

Response 3: This paragraph has been rewritten.

“Previous research works have thoroughly investigated influence of chemical composition and microstructure resulting from different thermal treatments on hot cracking susceptibility of alloy 718 [9][14][15][16][17]. This has led to contradictory conclusions about the influence of grain size on hot cracking susceptibility of alloy 718 castings. Whereas some authors have shown that coarser base metal grain sizes enhance FZ solidification cracking [17][18][19], others concluded on reduced cracking susceptibility of samples with coarser grains [15]”.

In response 2, you listed, that Introduction is good in your opinion. However, in response 3, you wrote, that you have changed it. You have exceed some information in this section, co you did, what I pointed in first round of review. Now it looks better, potential readers don't have to find in other references, what you mean.

Response 4: We used Mendeley pluging to insert and track references. We can edit the reference style, but cannot do consolidate references into one bracket. This is something that we can do during final editing.

I agree.

Response 5: We did not perform chemical analysis by ourselves. We included chemical composition of wrought material given in the material certificate (as standard AMS 5596) and the chemical composition of the ingots employed for the casting.

In previous state of the manuscript there wasn't info in this field. Now it is clear.

Response 6: By this sentence we tried to point out that we used always the same welding speed, arc length and electrode in our TIG Varestraint tests. 3 samples were tested for each augmented strain level (tests were carried out at 9 different levels) using both pulsed and continuous welding curves.

In previous state of the manuscript there wasn't info in this field. Now it is clear.

Response 7: Varestraint tests provides a good relative measurement of hot solidification cracking susceptibility. Liquation and solidification cracking, both of them contributing to hot solidification, can be studied separately in simple geometries and without sample preparation. This test allows to compare and rank cracking susceptibility of alloys (base metals and filler metals) and analyse influence of welding conditions on cracking. These advantages have been included in the new version.

For me it was clear, but should be also for potential readers. You have added this part, so now it looks good.

Response 8: We consider this comment a suggestion. We prefer to keep current structure which is usually requested for scientific papers. In our opinion, Results sections must describe the results that have been obtained and Discussion must explain these results linking with underpinning mechanism, hypothesis and previous findings (references). We think that the quality of the paper is higher keeping both sections as they are now, as far as good traceability between figures and tables is provided.

I don't agree with you. From my experience, articles with connected "results and discussion" looks better, have more citations and a lot of valuable journals allows to connect these parts. However, it was only my suggestion, so it could be separate.

Best regards.